# From Quantifying to Reducing Uncertainty: Diffusion Hypernetworks for Robust Medical Image Reconstruction

## Abstract

Accelerated medical imaging is widely used for reduced scan time and exposure to radiation, improving patient experience. However, sparse-view CT and accelerated MRI produce reconstructions that suffer from both aleatoric (acquisition noises, undersampling, patient motions) and epistemic (model uncertainty) variability. Prior work has focused on quantifying uncertainty, but reporting it alone does not improve the robustness of reconstructed images. We introduce a diffusion-based reconstruction framework with a Bayesian hypernetwork that explicitly reduces uncertainty rather than merely estimating it. Two complementary learning objectives target the distinct sources: noise-consistency to reduce aleatoric uncertainty and weight-consistency to reduce epistemic uncertainty. Trained in separate phases to avoid interference, these learning objectives produce reconstructions that are both high-quality and reliable. Experiments on sparse-view CT (LUNA16) and accelerated MRI (fastMRI Knee and Brain) show substantial reductions in both uncertainty components without degrading image quality, and consistent gains in downstream lung nodule segmentation and pathology classification performance. By shifting uncertainty from a diagnostic overlay to an optimization target, our method produces reconstructions that are anatomically accurate and clinically useful, advancing uncertainty-aware generative modeling for medical imaging.

## 1 Introduction

Accelerated medical imaging, such as sparse-view CT and accelerated MRI, is widely used in medical setting to reduce scan and exposure to radiation (CT only), improving patient comfort and safety. It relies on AI-based reconstruction algorithms to produce high-quality images. AI models are inherently uncertain, especially with low data input. Small perturbations in acquisition or model weights can yield images that look realistic yet unreliable, undermining downstream tasks like tissue segmentation and disease diagnosis.

Uncertainty in medical imaging reconstruction arises from two distinct sources. Aleatoric uncertainty (AU) reflects acquisition variability, including noises, sampling patterns, and patient motions. In contrast, epistemic uncertainty (EU) arises from the AI reconstruction model limitations, such as parameter ambiguity or incomplete training coverage. Prior work has primarily focused on quantifying AI models' (not necessarily medical image reconstruction models) uncertainties, — using Bayesian approximations (Kendall & Gal, 2017; Schlemper et al., 2018; Wu et al., 2021; Zhang et al., 2023), ensembles (Lakshminarayanan et al., 2016; Kuestner et al., 2024; Mehrtash et al., 2020), or Bayesian hypernetworks (BHNs) (Krueger et al., 2017) to quantify uncertainties. While useful, uncertainty quantification alone does not improve the underlying reconstructions. Radiologists and downstream AI tasks are still forced to use images that are degraded.

We believe that medical image reconstruction using AI will benefit not just from measuring uncertainty but from reducing it. We introduce a reconstruction framework with a diffusion model and a Bayesian hypernetwork. The framework explicitly targets reducing both uncertainty types. We propose two objectives to enforce clinically meaningful invariances:

- Noise-consistency: reconstructions remain stable under perturbed measurements, reducing aleatoric uncertainty.

- Weight-consistency: reconstructions agree across sampled parameter sets, reducing epistemic uncertainty.

We train these objectives in separate phases to avoid interference, yielding reconstructions that are both anatomically accurate and robust to noises and parameter variability. Evaluations on sparse-view CT (LUNA16) and accelerated MRI (fastMRI Knee and Brain) demonstrate that reducing uncertainty substantially improves reconstruction stability while preserving image quality. More importantly, these gains translate into higher performance on downstream tasks, including lung nodule segmentation as well as knee and brain pathology classification.

In summary, our contributions are:

1. A reconstruction framework based on a diffusion model and a Bayesian hypernetwork, which reduces rather than only estimates uncertainty in medical image reconstruction.

2. Novel training losses for aleatoric and epistemic uncertainty reduction, aligned with measurement and model invariances.

3. Empirical evidence across CT and MRI showing improved reconstruction quality and consistent benefits for downstream clinical AI tasks.

## 2 RELATED WORK

This research connects the studies on uncertainty quantification and reduction in the medical domain. We briefly review each topic below.

### 2.1 MEDICAL UNCERTAINTY QUANTIFICATION

Effective quantification of uncertainty is crucial for reliable and transparent medical decision- making, particularly in the context of diagnostic procedures. In the medical domain, commonly utilized approaches for uncertainty quantification are Bayesian inference (Li et al., 2021; Lin et al., 2020; Zhou et al., 2020; Akkoyun et al., 2020), Monte Carlo simulation (Silva et al., 2023; Salgado et al., 2020; Tsai et al., 2020), fuzzy systems (Castellazzi et al., 2020; Das et al., 2020; Liu et al., 2018), Dempster-Shafer's theory (Liu et al., 2023; Razi et al., 2019), rough set theory (Li et al., 2023; Acharjya et al., 2020; Santra et al., 2020), and imprecise probability (Giustinelli et al., 2022; McKenna et al., 2018; Mahmoud, 2016).

With the wide adoption of deep learning models in healthcare, the uncertainty quantification approaches focus more on deep learning contexts. Researchers in this space have extensively investigated four main approaches to quantify both data (aleatoric) and model (epistemic) uncertainties: 1) single deterministic methods (McKinley et al., 2021; Luo et al., 2020); 2) ensemble methods (McClure et al., 2018; Liang et al., 2020; Linmans et al., 2020); 3) test-time augmentation (Zhang et al., 2019; Athanasiadis et al., 2020; Ayhan et al., 2020); and 4) Bayesian methods (Kendall & Gal, 2017; Chan et al., 2024).

In this study, we adopted a Bayesian method, specifically, HyperDM (Chan et al., 2024), to quantify uncertainty for its effectiveness and computational efficiency. Our focus is to reduce uncertainty in addition to just quantifying it. We will introduce HyperDM more in the Methodology section.

### 2.2 MEDICAL UNCERTAINTY REDUCTION

Clinical machine learning models reduce uncertainty in two main ways: 1) during the training process and 2) during the inference process. During training, a common strategy is to make uncertainty part of the learning objective, e.g., down-weighting over-confident samples (Dawood et al., 2023) or contrastive learning (Jarimijafarbigloo et al., 2024). Another widely used strategy is active learning, which selects the most informative samples during training to reduce epistemic uncertainty (e.g. (Nath et al., 2020; Huang et al., 2024)). During inference, uncertainty-guided acquisition chooses the next k-space line in MRI using a trained model's uncertainty, thereby reducing

reconstruction error and uncertainty (Zhang et al., 2019). Prior works largely focus on epistemic uncertainty and rarely disentangles aleatoric and epistemic uncertainty.

In this work, we address that gap by explicitly separating and reducing aleatoric and epistemic uncertainty. Also, we treat uncertainty as a direct optimization target, rather than a weighting mechanism, to guide the image reconstruction.

# 3 METHODOLOGY

## 3.1 PRELIMINARIES ON HYPERDM (CHAN ET AL., 2024)

Our approach is inspired by the recent study, HyperDM (Chan et al., 2024), which estimates epistemic and aleatoric uncertainty with a single model. We advance it by reducing both. HyperDM (Chan et al., 2024) is based on a Bayesian hypernetwork (Krueger et al., 2017). Hypernetworks (Ha et al., 2016) employ a paradigm where one network, the "hyper" network generates weights for another "primary" network which performs the specific task. Bayesian hypernetworks (BHNs) (Krueger et al., 2017) extend hypernetworks to quantify uncertainty. Rather than accepting task-specific tokens as inputs, BHNs accept random noise and stochastically generate weights for the primary network. The primary network is a Diffusion Model (DM) (Ho et al., 2020), a type of deep generative model that utilizes the principles of diffusion and denoising processes to generate images. The paring of a DM and a BHN in a single model approximates the behavior of a deep ensemble model without training many separate networks.

During training, the BHN $h_\phi$ maps a low-dimensional noise $z$, drawn from a multivariate standard normal distribution, $z \sim \mathcal{N}(\mathbf{0}, \mathbf{I})$, to a plausible set of DM's weights $\theta(z)$. In effect, a single BHN replaces many separately trained DMs and keeps sampled weights within the range of valid diffusion models. At inference time, HyperDM draws $z \sim \mathcal{N}(\mathbf{0}, \mathbf{I})$ and pass it through the hypernetwork to obtain a set of weights $\theta(z)$ for the DM $f_{\theta(z)}$. Repeating this $M$ times produces an implicit posterior over weights that captures epistemic uncertainty.

For a fixed weight set $\theta(z)$ and an input condition $y$, the DM is run $N$ times with different denoising trajectories. Each trajectory starts from a Gaussian noise $x_T$ and iteratively predicts the added noise $\epsilon_\mathbf{t}$ at every timestep to generate a clean image $\hat{x} \sim q(x|y, \theta(z))$. Variation across these $N$ denoising trajectories reflects aleatoric uncertainty.

After collecting the $M \times N$ predictions (generated images) $\{x^{(i,j)}\}_{M \times N}$, HyperDM computes the total variance:

$$\underbrace{\text{Var}(\hat{x})}_{\text{Total Uncertainty}} = \underbrace{\text{Var}_{i \in M}\Big[\mathbb{E}_{j \in N}\big[\hat{x}^{(i,j)}\big]\Big]}_{\text{EU}} + \underbrace{\mathbb{E}_{i \in M}\Big[\text{Var}_{j \in N}\big[\hat{x}^{(i,j)}\big]\Big]}_{\text{AU}}. \tag{1}$$

The first term captures the uncertainty given by the variance of $M$ sampled weights $\theta(z)$ over the $N$ expected values of $\hat{x}^{(i,j)}$. This term ignores variance caused by the data-inherent randomness, and therefore represents EU. The second term captures the uncertainty given by the expectation of $M$ sampled weights $\theta(z)$ over the variance of $N$ $\hat{x}^{(i,j)}$. This term ignores the variance caused by the sampling of weights and therefore represents AU.

The generated image is the ensemble mean: $\bar{x} = \frac{1}{M \times N} \sum_{i,j} \hat{x}^{(i,j)}$. It is worthwhile to note that only the BHN's parameters, $\phi$, are trained. The DM's parameters are purely generated, $\theta(z) = h_\phi(z)$.

## 3.2 PROPOSED APPROACH: REDUCING ALEATORIC AND EPISTEMIC UNCERTAINTY

We will advance the uncertainty research by reducing both aleatoric and epistemic uncertainty, in addition to merely quantifying them.

### 3.2.1 REDUCING ALEATORIC UNCERTAINTY

Aleatoric uncertainty (AU) arises from data-inherent noises and randomness, such as scanner artifacts, patient motions, or projection variability. We will reduce AU by training the reconstruction model to be insensitive to these noises and randomness.

Specifically, during training, for each mini-batch, we sample $N$ independent noisy variants of the input condition $y$ by adding independent and identically distributed (i.i.d.) zero-mean Gaussian noise $\epsilon_{\mathbf{i}} \sim \mathcal{N}(\mathbf{0}, \sigma^2 \mathbf{I})$. Each noisy condition $y_i$ is passed through the DM, producing a version of reconstruction. If the model were robust to noises and randomness, the series of reconstructions would be nearly identical. The process could be mathematically described as:

$$y_i = y + \epsilon_{\mathbf{i}}, \qquad \epsilon_{\mathbf{i}} \sim \mathcal{N}(\mathbf{0}, \sigma^2 \mathbf{I}), \tag{2}$$

$$\hat{x}_i = f_{\theta(z)}\big(x_T, y_i\big), \qquad x_T \sim \mathcal{N}(\mathbf{0}, \mathbf{I}). \tag{3}$$

To isolate the problem of AU reduction, we keep the *weights the same* across $y_i$ (no weight sampling for this objective), and we fix $x_T$ for the $N$ denoising trajectories. That way, any variance of $\hat{x}_i$ is attributed only to the input condition $y_i$.

For the training loss function, in addition to the original reconstruction loss, we add a loss term to penalize the variance of these $N$ reconstructions induced by the noisy conditions, driving the model toward aleatorically robust predictions. Therefore, the overall loss function for the BHN is:

$$\mathcal{L}_{\text{BHN-AU}} = \frac{1}{D} \sum_{(x,y) \in D} \big[ |\hat{x} - x| + \lambda_{\text{AU}} \text{Var}_{i \in N}\big[\hat{x}_i\big] \big], \tag{4}$$

where $\hat{x} = f_{\theta(z)}(x_T, y)$ is the reconstruction on the clean input condition $y$, and $x$ is the ground truth reconstruction. The first term enforces the predictions $\hat{x}$ to approximate the ground truth $x$, preventing trivial smoothing; the second enforces *noise-consistency*, shrinking the empirical variance of reconstructions induced by perturbations of $y$. Intuitively, if $f_\theta$ is robust to acquisition noises and randomness, $\{\hat{x}_1, \ldots, \hat{x}_N\}$ should agree up to negligible residuals. Thus minimizing $\text{Var}_{i \in N}\big[\hat{x}_i\big]$ directly targets AU by suppressing the sensitivity to acquisition noises and randomness. The $\lambda_{\text{AU}}$ trades off the reconstruction fidelity and sensitivity to noises. We optimize only the BHN parameters, keeping the base DM's parameters fixed.

### 3.2.2 REDUCING EPISTEMIC UNCERTAINTY

Epistemic uncertainty arises from incomplete knowledge of the model parameters given finite training data. For reconstruction tasks, training a single set of weights $\theta$ typically converges to one most-likely weight settings, leaving other plausible solutions unexplored and making reconstructions overconfident. Our goal is not merely to average over weight samples, but to shape the BHN's weight distribution so that different plausible parameter draws agree on their reconstructions, i.e., to concentrate the posterior on solutions that are both accurate and mutually consistent.

For a condition $y$ with ground truth $x$, we draw $M$ sets of weights via BHN:

$$z^{(m)} \sim \mathcal{N}(\mathbf{0}, \mathbf{I}), \qquad \theta(z^{(m)}) = h_\phi\Big(z^{(m)}\Big), \qquad m = 1, \ldots, M, \tag{5}$$

then perform the reconstruction task with the DM:

$$\hat{x}^{(m)} = f_{\theta(z^{(m)})}(x_T, y), \tag{6}$$

where we fix $x_T$ across the $M$ times so that variations among $\{\hat{x}^{(m)}\}$ reflect the *weight* variability.

We combine the reconstruction loss term with a *weight consistency* term:

$$\mathcal{L}_{\text{BHN-EU}} = \frac{1}{D} \sum_{(x,y) \in D} \big[ |\bar{x} - x| + \lambda_{\text{EU}} \frac{1}{\binom{M}{2}} \sum_{1 \le i < j \le M} |\hat{x}^{(i)} - \hat{x}^{(j)}| \big]. \tag{7}$$

The first term is the standard reconstruction loss evaluated on the mean prediction across $M$ sets of weights where $\bar{x} = \frac{1}{M} \sum_{m=1}^M \hat{x}^{(m)}$. The second term minimizes average pairwise deviation, a

robust proxy for posterior spread over reconstructions; driving it down encourages independently sampled weights to agree, thereby concentrating the induced weight posterior around high-quality solution. The coefficient $\lambda_{\mathrm{EU}}$ trades off the reconstruction fidelity and weight consensus. We optimize only the BHN parameters, keeping the DM fixed.

We train $\mathcal{L}_{\mathrm{BHN-EU}}$ and $\mathcal{L}_{\mathrm{BHN-AU}}$ as two separate objectives in independent runs (AU-only and EU-only), with no parameter sharing between them. The models are named **AUDiff** and **EUDiff** respectively. This avoids loss-scale imbalance and gradient conflict between input noise invariance (AU) and weight invariance (EU), and cleanly attributes each objective's contribution.

## 4 EXPERIMENTS

### 4.1 DATASETS AND TASKS

To evaluate the performance of the noise-consistency and weigh-consistency learning objectives, we select three publicly available datasets: one CT dataset and two MRI datasets. A summary of dataset statistics is reported in Table 1.

- **LUNA16 (Setio et al., 2017):** A subset of the LIDC–IDRI archive (Lung Image Database Consortium and Image Database Resource Initiative) containing 888 chest CT scans. Each scan is a 3D volumetric image that we treat as a stack of axial 2D slices. It is an annotated dataset for lung nodule segmentation (binary masks).
- **fastMRI Knee:** The knee subset contains fully sampled clinical knee MRIs acquired with single-coil and multi-coil scanners. Each MRI scan is provided as a 3D image. We convert each 3D image into an axial stack of 2D slices. We use the fastMRI (Zbontar et al., 2018) dataset together with fastMRI+ (Zhao et al., 2022), an annotation extension that provides 22 slice-level pathology labels for knees (e.g., meniscus tear, joint effusion, ligament - ACL high grade sprain, etc.).
- **fastMRI Brain:** The brain subset consists of fully sampled brain MRI scans acquired predominantly using multi-coil MRI scanners. Similar to the fastMRI Knee dataset, each MRI scan is a 3D image and we convert each 3D image into an axial stack of 2D slices. We also use it with fastMRI+ annotations with 30 slice-level pathology labels for brains (e.g., likely cysts, mass, lacunar infarct, etc.).

Table 1: Dataset statistics

| Dataset | No. of 3D Scans | No. of Derived 2D Slices |
|---|---|---|
| LUNA16 | 888 | 227,225 |
| fastMRI Knee | 1,594 | 49,779 |
| fastMRI Brain | 6,970 | 117,596 |

### 4.2 INPUT CONDITIONS FOR THE DM AND THE RECONSTRUCTION DETAILS

Diffusion models need input conditions to steer the denoising process to generate the desired reconstructed images. For CT, following most AI reconstruction models, we use the sinograms to be the input conditions. A sinogram in CT is the 2D representation of X-ray projections from many angles collected as the CT scanner rotates. It's the raw data used to reconstruct the CT slice. However, such sinograms are not available in the LUNA16 dataset. Therefore, for each CT slice, we simulated a sparse-view (45-view equiangular 0-360) sinogram through a forward Radon transform process. The reconstruction model (DM) will take the sparse-view sinogram as the input condition and generate a full-view CT slice.

For MRI, deep learning models often take zero-filled images as input and learn to "de-alias" them to reconstruct higher-quality images. Therefore, we will obtain the zero-filled images from the fastMRI dataset as the input conditions of the DM. Specifically, we under-sampled k-space lines and insert zeros in the unsampled portion of k-space. We then applied the inverse Fourier transform to obtain the zero-filled image as the condition.

We use a 2D U-Net denoiser in the DM with sinusoidal time embeddings trained with $T = 1000$ linear $\beta$ steps, and a lightweight BHN that maps $z \sim \mathcal{N}(\mathbf{0}, \mathbf{I})$ to the U-Net weights to enable weight

sampling. During training, we present each slice with $N = 5$ perturbed conditions and sample $M = 3$ weight sets per training batch. At inference, we draw $M = 10$ sets of weights and, for each set, $N = 100$ stochastic trajectories for each inference sample. The mean gives the reconstructed image, the variance across the trajectories (under the fixed weights) yields the AU map (a matrix as the second component in Equation 1), and the variance across the weights (under the fixed condition) yields the EU map (a matrix as in the first component in Equation 1).

### 4.3 Evaluation Metrics on the Reconstruction Tasks

We would like to evaluate the reconstruction performance from two perspectives: how much uncertainty (both AU and EU) is reduced and whether the reconstructed images remain high-quality. For the AU value, we will average the values of all the AU matrix (map) elements. For the EU value, we will do the same for the EU matrix (map). By comparing AU and EU values before and after our noise-consistency and weight-consistency objectives, we are able to show exactly how much each type of uncertainty is reduced.

To ensure uncertainty reduction does not wash out clinically relevant details, we report the quality of the reconstructed images using two established metrics: Peak Signal-to-Noise Ratio (PSNR) (Hore & Ziou, 2010) and Structural Similarity Index (SSIM) (Wang et al., 2004). A higher PSNR (typically ranges in $[0, \infty)$) means the reconstruction is closer, pixel-by-pixel, to the original image. A higher SSIM (typically ranges in $[0, 1]$) indicates the similarity of the reconstructed image to the original image in terms of contrast, textures, and edges. Together, PSNR and SSIM verify that reducing uncertainty does not come at the expense of visual quality and anatomical detail of medical images.

### 4.4 Further Evaluations of the Reconstructed Images: Downstream Medical Segmentation and Classification Tasks

After evaluating the reconstructed images in terms of AU, EU, PSNR, and SSIM, we still have three questions: (1) do these reconstructions preserve diagnostic content learned from original images? (2) are reconstructions viable when only reconstructed images are available (without the original ground truth)? and (3) do reconstructed images supplemented with corresponding AU and EU maps help in the downstream medical tasks? In answering these questions, we apply these images for the downstream clinical tasks: lung nodule segmentation for the CT reconstructions and pathology classification for the MRI reconstructions. For each reconstruction model, HyperDM (Chan et al., 2024) (the base model), AUDiff, and EUDiff, we generate one reconstruction per slice along with its AU and EU maps (two matrices), then use them to train and test the downstream task models.

We used three training and test settings for the downstream tasks to mirror three real-life scenarios:

Setting 1: Train on the originals but test on the reconstructions: train the task model on scanner-acquired original slices (source distribution) and evaluate it on slices produced by each reconstruction model (target distribution) to test the construction model's generalizability. The two sets differ in intensity statistics, noise or artifacts patterns, and edge sharpness. This mirrors the scenarios where clinicians often train AI on regular images but deploy on accelerated images (sparse-view CTs, accelerated MRIs, etc.). If performance drops, reconstructions are missing task-relevant signals.

Setting 2: Train and test both on the reconstructions: This mirrors the situation when only reconstructed images are available. Any performance differences reflect the intrinsic quality of the reconstruction model itself.

Setting 3: Train and test both on the uncertainty-augmented reconstructions: Train and test on reconstructions augmented with the AU and EU maps, which are concatenated as additional input channels. This evaluates whether localized uncertainty cues provide useful signals, allowing the task model to prioritize reliable pixels and down-weight ambiguous regions.

For the CT segmentation task, we train a U-Net with the Dice and binary cross-entropy loss. For the MRI multi-label classification tasks, we train a ResNet-34 model with a sigmoid multi-label head and binary cross-entropy loss. We evaluate the CT segmentation task using the Dice score, which quantifies the overlap between the predicted masks and the ground truth segmentation. The score ranges from 0 (no overlap) to 1 (perfect overlap), with higher values indicating larger overlaps. For

the MRI multi-label classification tasks, we report classification accuracy as well as macro ROC-AUC (abbreviated as AUC in the Results section) for better understanding performance on rare pathology labels in the class imbalance situation. We split the data into training (70%), validation (10%), and test (20%) sets to train and evaluate the downstream task model. All splits are strictly at the patient level and no patient appears in more than one partition.

# 5 RESULTS

## 5.1 UNCERTAINTY REDUCTION RESULTS

Table 2 reports the reconstruction AU and EU values along with the PSNR and SSIM values. Across all datasets, the proposed objectives drastically reduce uncertainty relative to HyperDM. AUDiff primarily enhances robustness to input noise. EUDiff enforces structural consistency across weight samples, producing the lowest EU and AU (as a side effect) and the highest SSIM. Together, these results demonstrate that uncertainty reduction can be achieved without compromising, and often improving the reconstruction fidelity.

It is interesting to notice from Table 2 that the noise-consistency (AUDiff) objective reduces EU as a side effect, even though it was designed to target AU. The reason could be making reconstructions robust to acquisition noise restricts the weight posterior to agreed-on solutions. Notably, weight-consistency (EUDiff), designed to reduce EU, suppresses AU more than AUDiff. The reason could be enforcing agreement across weight samples encourages the hypernetwork to generate weights that produce consistent internal representations. These stable representations filter out random acquisition noises more effectively. In summary, weight-driven consistency provides most powerful reduction in both EU and AU. In addition, the two objectives are not strictly independent, so regularizing one source of variability helps the other.

Table 2: Reconstruction uncertainty and quality across the datasets and models. $\uparrow$ means the larger the better and $\downarrow$ means the smaller the better. The best result in each column is bolded.

| Dataset | Model | AU$\downarrow$ | EU$\downarrow$ | PSNR$\uparrow$ | SSIM$\uparrow$ |
|---|---|---|---|---|---|
| LUNA16 | HyperDM | $2.15\times10^{-3}$ | $2.00\times10^{-4}$ | 39.15 | 0.9277 |
| | AUDiff | $7.98\times10^{-9}$ | $2.02\times10^{-6}$ | **40.90** | 0.9126 |
| | EUDiff | $\mathbf{1.02 \times 10^{-11}}$ | $\mathbf{4.83 \times 10^{-12}}$ | 39.89 | **0.9911** |
| fastMRI Knee | HyperDM | 0.0391 | 0.0130 | 9.81 | 0.3410 |
| | AUDiff | 0.0040 | 0.0005 | 11.57 | 0.3620 |
| | EUDiff | **0.0021** | **0.0001** | **13.09** | **0.3710** |
| fastMRI Brain | HyperDM | 0.0140 | 0.0040 | **13.61** | 0.5350 |
| | AUDiff | 0.0009 | 0.0004 | 13.18 | 0.5320 |
| | EUDiff | **0.0007** | **0.0000** | 13.41 | **0.5360** |

## 5.2 DOWNSTREAM MEDICAL SEGMENTATION AND CLASSIFICATION

We next evaluate whether uncertainty-reduced reconstructed images improve downstream medical tasks. Specifically, we assess lung nodule segmentation on LUNA16 and pathology classification on fastMRI Knee and Brain. Results are analyzed under three distinct training settings. The segmentation results are shown in Table 3 and the classification results are shown in Table 4.

Setting 1: Train on the originals but test on the reconstructions. This setting exposes distribution shift: task models trained on regular medical images (originals) should generalize to accelerated images (reconstructions). We find that AUDiff and EUDiff both have better performance in both tasks than HyperDM, confirming that HyperDM reconstructed images are more different from the originals, obscuring task-relevant features. By contrast, uncertainty-aware models substantially mitigate this distribution shift and therefore performance degradation. Both noise-consistency and weight-consistency objectives suppressed AU/EU by several orders of magnitude. These results indicate

Table 3: The LUNA16 CT lung nodule segmentation results under three training and test settings. Dice is reported; higher value indicate better performance. The best result in each row is bolded.

| Training and Test Setting | HyperDM | AUDiff | EUDiff |
|---|---|---|---|
| Setting 1 | 0.6725 | 0.7976 | **0.8319** |
| Setting 2 | 0.7758 | 0.7914 | **0.8008** |
| Setting 3 | 0.7914 | 0.8257 | **0.8331** |

Table 4: The fastMRI Knee and Brain multi-label pathology classification results under three training and test settings. Accuracy and AUC are reported for all settings. For both metrics, higher values are better. The best result in each row is bolded.

| Dataset | Training and Test Setting | HyperDM | | AUDiff | | EUDiff | |
|---|---|---|---|---|---|---|---|
| | | Accuracy | AUC | Accuracy | AUC | Accuracy | AUC |
| fastMRI Knee | Setting 1 | 0.4285 | 0.8217 | 0.4462 | **0.8311** | **0.4471** | 0.8298 |
| | Setting 2 | 0.4314 | 0.8774 | 0.4853 | 0.9008 | **0.4882** | **0.9138** |
| | Setting 3 | 0.4769 | 0.8821 | 0.5098 | 0.9179 | **0.5125** | **0.9251** |
| fastMRI Brain | Setting 1 | 0.5255 | 0.8589 | 0.5741 | **0.8620** | **0.5749** | 0.8608 |
| | Setting 2 | 0.5412 | 0.8608 | 0.5878 | 0.9038 | **0.5887** | **0.9108** |
| | Setting 3 | 0.5496 | 0.8889 | 0.5953 | 0.9139 | **0.6015** | **0.9185** |

that stable, low-uncertainty reconstructions transfer more reliably across the originals and reconstructions.

**Setting 2: Train and test both on the reconstructions.** Here the distribution mismatch is removed, and task performance reflects the inherent quality of the reconstructions themselves. Again, uncertainty-aware training provides advantages over HyperDM. EUDiff's reduced EU fosters consistent structural representations, leading to higher Dice and classification accuracy. Thus, training on reconstructions amplified by uncertainty reduction produces cleaner supervision signals and steadier task learning.

**Setting 3: Train and test both on the uncertainty augmented reconstructions.** Finally, we test whether explicitly supplying AU and EU maps as additional channels aids the task models. This setting yields the best outcomes. The uncertainty maps serve as spatial reliability cues, allowing the downstream models to emphasize stable anatomical regions and down-weight ambiguous boundaries. Notably, EUDiff consistently outperforms AUDiff, reflecting its sharper reduction of EU (Table 2) and suggesting that weight-driven consistency provides the most useful guidance.

Across all three training settings, reconstructions with reduced uncertainty lead to measurable gains in downstream CT segmentation and MRI classification accuracy. Improvements are most pronounced when uncertainty maps are leveraged explicitly, highlighting their value as reliability-aware features. Collectively, these findings demonstrate that uncertainty reduction not only stabilizes image reconstruction but also enhances clinical task performance under realistic deployment scenarios.

### 5.3 HYPERPARAMETER ANALYSIS

We performed separate searches for the weights of $\lambda_{AU}$ and $\lambda_{EU}$ in the noise-consistency and weight-consistency objectives (Equations 4 and 7). For both weights, the searches are in the range of $\{0.00, 0.25, 0.50, 0.75, 1.00\}$. The result is shown in Figure 1. The best setting for $\lambda_{AU}$ is 0.75 for the lowest AU and EU (side effect) and highest PSNR and SSIM. The best setting for $\lambda_{EU}$ is 0.50 for lowest EU and AU (side effect) and highest PSNR and SSIM. Note that the settings of $\lambda_{AU} = 0$ and $\lambda_{EU} = 0$ serve as ablation studies that remove the corresponding AU or EU regularizer from the loss function. The removal of either leads to both AU/EU increases and PSNR/SSIM decreases. These findings validate the effectiveness of our reconstruction framework and underscore the impor-

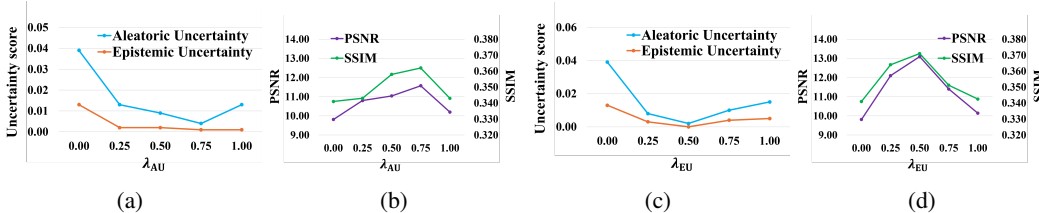

Figure 1: Hyperparameter sensitivity of the noise-consistency hyperparameter $\lambda_{AU}$ (a-b) and the weight-consistency hyperparameter $\lambda_{EU}$ (c-d) for fastMRI Knee. Each curve in (a) and (c) reports the test-set mean per-pixel AU (blue) and EU (orange) values. The lower the better. Each curve reports in (b) and (d) reports the test-set mean of PSNR (purple) and SSIM (green). The higher the better.

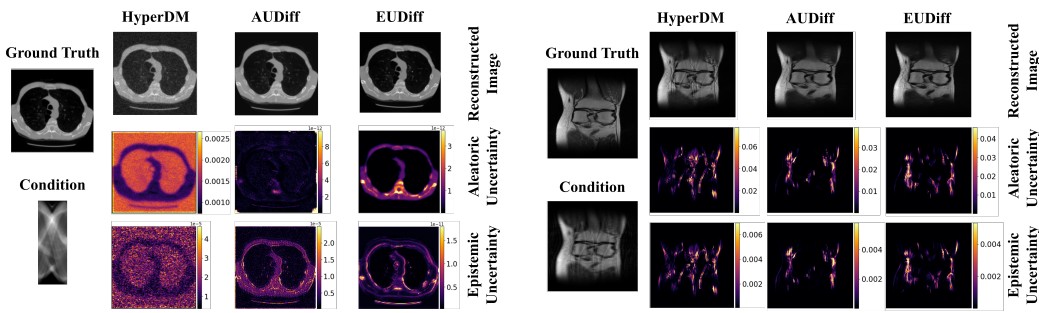

Figure 2: Two case studies. Left group: a slice in LUNA16; right group: a slice in fastMRI Knee. The leftmost column shows the input condition and the ground truth image. The remaining columns display resulting images by HyperDM, AUDiff, and EUDiff. The first row shows the reconstructed images, the second and third rows show the corresponding AU and EU maps. Brighter colors indicate higher uncertainty values in the maps.

tance of balanced uncertainty regularization in achieving high-quality reconstructions with minimal uncertainty.

### 5.4 CASE STUDY

To illustrate how uncertainty reduction translates into both visual and interpretive improvements, Figure 2 compares reconstructions from HyperDM, AUDiff, and EUDiff on one representative CT slice and one representative MRI slice, along with their corresponding AU and EU maps.

HyperDM reconstructions appear blurrier, with the uncertainty maps that highlight broad, non-specific regions. In contrast, AUDiff produces sharper images with noise artifacts largely suppressed, and AU maps that contract from widespread coverage to narrow bands around edges and areas prone to patient motions (breathing, heartbeat, etc). EUDiff yields the cleanest and most stable reconstructions: boundaries are crisp, textures remain consistent, and both AU and EU maps are tightly localized to the most challenging structures, such as thin tissue interfaces or undersampled regions.

### 6 CONCLUSION

We present a diffusion-based reconstruction framework with Bayesian hypernetworks that explicitly reduces both aleatoric and epistemic uncertainty through noise- and weight-consistency objectives. Across CT and MRI benchmarks, our approach substantially lowered uncertainty, improved reconstruction quality, and boosted downstream clinical performance in segmentation and classification. By treating uncertainty as an optimization target rather than just a diagnostic overlay, our method delivers reconstructions that are both anatomically accurate and clinically useful, advancing uncertainty-aware generative modeling toward reliable real-world deployment.

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
