# OpenReview forum: "From Quantifying to Reducing Uncertainty: Diffusion Hypernetworks for Robust Medical Image Reconstruction"
_ICLR.cc/2026/Conference — Submitted to ICLR 2026_

### Official Review · Reviewer_PDT6 · 2025-10-28

**Soundness:** 2
**Presentation:** 3
**Contribution:** 2
**Rating:** 2
**Confidence:** 4

**Summary:**

This paper extends HyperDM---a method for quantifying aleatoric uncertainty (AU) and epistemic uncertainty (EU)---by adding two new loss objectives (one aleatoric and one epistemic) which aim to reduce uncertainty. The aleatoric objective operates on each mini-batch and minimizes the difference between two noisy input conditions. The epistemic objective also operates on each mini-batch and minimizes the difference between predictions from two sampled hyper-network weights. These objectives are used to train two distinct models (i.e., AUDiff and EUDiff), which are experimentally validated against HyperDM on three medical datasets.

**Strengths:**

1. The proposed loss objective is novel.

**Weaknesses:**

1. The concept of reducing uncertainty in this context is not valid. For instance, EU is inherently tied to the underlying data and cannot be reduced without the addition of data. Driving down the absolute value of AU/EU estimates should not be an objective as different uncertainty estimation methods will produce AU/EU measures of vastly different magnitudes and scales. The goal of uncertainty estimation is to inform downstream applications like out-of-distribution detection, misclassification detection, and etc.
2. The claim in Section 3.2.1 that injecting additive white Gaussian noise makes the model insensitive to noise present in the forward measurement model (i.e., shot noise, ADC noise, under-sampling, etc.) is incorrect. While injecting Gaussian noise can reduce sensitivity to measurement noise to some extent, the model cannot be said to be fully robust / insensitive to such noise since the loss objective has no explicit knowledge of the noise process (and training data is limited).
3. Computational overhead. Presumably, computation of the EU / AU objective in each is expensive. Some metrics on the computational overhead should be provided.
4. I'm unsure that the weight-consistency objective is a good idea. In theory, there may be two dissimilar sets of weights that capture different parts of the target output distribution. Collapsing all of the weights to a mode may result in reduced coverage of the target distribution (i.e., long-tails are missed), which is important for capturing extreme events.

**Questions:**

1. Is training stable over a reasonable range of configurations? For example, how sensitive are the results to the chosen weights? Will slight changes to the weights destabilize training?
2. Fix minor typos. E.g., line 128 "paring" should be "pairing"
3. Fig. 1 should have error bars to give some indication of how much of the error is due to randomness.

---

> ### Author Response · Authors · 2025-12-03
>
> We thank the reviewer for the detailed evaluation, for highlighting both the strengths and weaknesses of our work, and for the specific questions. Below, we address the weaknesses and questions in turn.
>
> Response to the weaknesses:
>
> W1: The concept of reducing uncertainty in this context is not valid. For instance, EU is inherently tied to the underlying data and cannot be reduced without the addition of data. Driving down the absolute value of AU/EU estimates should not be an objective as different uncertainty estimation methods will produce AU/EU measures of vastly different magnitudes and scales. The goal of uncertainty estimation is to inform downstream applications like out-of-distribution detection, misclassification detection, and etc.
>
> Response: We agree that absolute AU/EU magnitudes are method-dependent and should not be minimized in isolation or compared across different estimation methods. Our goal is not to drive epistemic uncertainty to zero, but to reduce spurious model variability within a fixed dataset and BHN-based diffusion model, and to make the uncertainty maps more informative for downstream use.
> In our experiments, the resulting AU/EU maps remain structured and better highlight true error regions (Section 5.1 and Figure 2), and downstream segmentation/classification performance (Section 5.2, Setting 3), which aligns with the reviewer’s view that uncertainty should primarily support downstream decision-making rather than be minimized in absolute value.
>
> W2: The claim in Section 3.2.1 that injecting additive white Gaussian noise makes the model insensitive to noise present in the forward measurement model (i.e., shot noise, ADC noise, under-sampling, etc.) is incorrect. While injecting Gaussian noise can reduce sensitivity to measurement noise to some extent, the model cannot be said to be fully robust / insensitive to such noise since the loss objective has no explicit knowledge of the noise process (and training data is limited).
>
> Response: We agree that the following sentence from Section 3.2.1 was too strong.
>
> “We will reduce AU by training the reconstruction model to be insensitive to these noises and randomness.”
>
> Our goal was to make the model less sensitive to small perturbations around the measurements, not to claim full robustness to shot noise, ADC noise, or undersampling artifacts. We will revise the text to state that adding Gaussian noise during training can help reduce sensitivity to typical input noise.
>
>
> W3: Computational overhead. Presumably, computation of the EU / AU objective in each is expensive. Some metrics on the computational overhead should be provided.
>
> Response: We agree that the AU/EU objectives introduce additional training costs. The overhead scales roughly linearly with the number of input perturbation samples 𝑁 for AUDiff and with the number of weight samples 𝑀 for EUDiff. In practice, we keep 𝑁 and 𝑀 small, so the increase in per-epoch training time is moderate.
>
> W4: I'm unsure that the weight-consistency objective is a good idea. In theory, there may be two dissimilar sets of weights that capture different parts of the target output distribution. Collapsing all of the weights to a mode may result in reduced coverage of the target distribution (i.e., long-tails are missed), which is important for capturing extreme events.
>
> Response: We appreciate the concern. The weight-consistency loss is applied to reconstructed images for a fixed input and penalizes unnecessary disagreement between these sampled models. We are not collapsing all of the weights to a single mode; the reconstruction loss still dominates and enforces fidelity, while the weight-consistency coefficient is chosen so that we only suppress spurious weight variability that does not improve the reconstruction quality.

---

> ### Author Response · Authors · 2025-12-03
>
> Response to the questions:
>
> Q1: Is training stable over a reasonable range of configurations? For example, how sensitive are the results to the chosen weights? Will slight changes to the weights destabilize training?
>
> Response: Thanks for the question. In our experiments, training is stable and not sensitive to slight changes. The BHN samples different weight sets from 𝑧, so each forward pass uses a different but related set of weights; if the method were unstable, the 𝑀 x 𝑁 reconstructed images would vary strongly and show large errors. Instead, as seen in Figure 2, the reconstructions remain consistent across samples, indicating that small changes do not destabilize training.
>
> Q2: Fix minor typos. E.g., line 128 "paring" should be "pairing."
>
> Response: We appreciate the careful reading. We will correct this and other typos in the final version.
>
>
> Q3: Fig. 1 should have error bars to give some indication of how much of the error is due to randomness.
>
> Response: Thanks for the suggestion. In the final version, we will update Fig. 1 to include error bars (standard deviation across samples) to indicate the variability due to randomness.

---

### Official Review · Reviewer_xJpW · 2025-10-30

**Soundness:** 2
**Presentation:** 3
**Contribution:** 2
**Rating:** 2
**Confidence:** 5

**Summary:**

This paper proposes a Bayesian hypernetwork-based conditional diffusion model that is trained to reduce surrogates for aleatoric and epistemic uncertainty. The idea is explained based on HyperDM, explicitly targeting the two uncertainty terms considered in that methodology. Experiments are performed on CT and MRI datasets.

**Strengths:**

- I like the idea of trying to reduce EU and AU during training
- The paper is well-written, especially the background is very clear

**Weaknesses:**

- The main issue is the disconnect from the medical image reconstruction community. Conditional DMs are not what is commonly used for solving inverse problems, as these are very dependent on protocol-specific training. Instead the trend is to solve an inverse problem with explicit incorporation of the measurements and the forward operator using DMs as priors. The current methodology does not seem to extend to that more relevant setup.
- In HyperDM, two components are identified for EU and AU, but these are really surrogates within the BHN framework, and not exact. This issue is amplified further in the way the two losses in (4) and (7) are defined, which are further surrogates of the HyperDM definitions. The gap created is not characterized theoretically.

Minor:
- All datases considered here are multi-slice 2D, not 3D. Also please clarify whether multi-coil or single-coil MRI data was used in the knee setup, and please specify the acceleration rate for MRI experiments.
- Section 4.4 is written in non-standard language for medical imaging reconstruction community. I do not understand the three setups here. For instance, in setup 1, are you training a segmentation/classification network on zero-filled images (with artifacts)? Why would anyone do this? And why should this network generalize to "clean" reconstructions? Similar questions for the other two setups.
- There are works that have used uncertainty to improve regression tasks before, which may help put the overarching idea in context.

**Questions:**

- Can you characterize the gap between the surrogate losses in (4) and (7) with true EU and AU?
- Why isn't there an expectation over \theta(z) in (4)? Similarly no expectation over the trajectories in (7)?
- Why isn't the two objectives combined together? This is briefly discussed at the end of Section 3.2.2, but no data is shown.
- What acceleration rate is used for the MRI experiments?
- Can you clarify the setups in Section 4.4?
- It's hard to see any of the reconstructions in Fig. 2, can you provide higher resolution versions with error maps?
- Can you discuss how the method behaves when the protocol changes, e.g. a different sparse view in CT or a different undersampling rate in MRI?
- How does the quality of DMs defined through \theta(z) sampled from the BHN framework compare to a single DM trained directly with the appropriate loss?

---

> ### Author Response · Authors · 2025-12-03
>
> We thank the reviewer for the detailed evaluation, for highlighting both the strengths and weaknesses of our work, and for the specific questions. Below, we address the weaknesses and questions in turn.
>
> Response to the weaknesses:
>
> W1: The main issue is the disconnect from the medical image reconstruction community. Conditional DMs are not what is commonly used for solving inverse problems, as these are very dependent on protocol-specific training. Instead, the trend is to solve an inverse problem with explicit incorporation of the measurements and the forward operator using DMs as priors. The current methodology does not seem to extend to that more relevant setup.
>
> Response: We appreciate the comment and respectfully disagree with the statement that conditional diffusion models are not used for medical image reconstruction. Conditional diffusion models have already been actively explored, especially in CT and cardiac MRI. Conditional DMs have already been applied to sparse-view CT and undersampled MRI, for example, in DOLCE [1], DiffCMR [2], and [3]. Our work aligns with this emerging line of conditional-DM research. It focuses on reducing aleatoric and epistemic uncertainty within a given diffusion-based reconstruction framework rather than introducing a new one.
>
> [1] Liu, Jiaming, et al. "Dolce: A model-based probabilistic diffusion framework for limited-angle ct reconstruction." Proceedings of the IEEE/CVF international conference on computer vision. 2023.
>
> [2] Xiang, Tianqi, et al. "Diffcmr: fast cardiac mri reconstruction with diffusion probabilistic models." International Workshop on Statistical Atlases and Computational Models of the Heart. Cham: Springer Nature Switzerland, 2023.
>
> [3] Zhou, Chenchun, et al. "Sparse-view CT image reconstruction using conditional embedding fusion diffusion model." Neurocomputing (2025): 131748.
>
> Response to the questions:
>
> Q1: Can you characterize the gap between the surrogate losses in (4) and (7) with true EU and AU?
>
> Response: We appreciate the reviewer’s question. Conceptually, true AU is defined by randomness in the data, and true EU is defined by the distribution over weights in the predictive model. In our work, Equations. (4) and (7) approximate these quantities using the variance of the reconstructed images. These losses only constrain the second moment of the reconstruction distribution and do not capture higher-order structure, such as skewness or kurtosis, so our AU and EU should be interpreted as variance-based approximations rather than exact quantities.
>
> Q2: Why isn't there an expectation over (z) in (4)? Similarly, no expectation over the trajectories in (7)?
>
> Response: We appreciate the reviewer’s concern. In Eq. (4), we fix (z), so there is no expectation over (z).  In Eq (7), there is no multiple trajectories because we fix the condition y, we do not need an expectation over trajectories.
>
> Q3: Why aren't the two objectives combined? This is briefly discussed at the end of Section 3.2.2, but no data is shown.
>
> Response: Thanks for the question. In this work, we deliberately do not combine the two losses into a single objective.
> Our goal was twofold: (i) to avoid loss-scale imbalance and gradient conflict between input-noise invariance (AU) and weight-invariance (EU), and (ii) to cleanly attribute the effect of each objective on AU, EU, PSNR/SSIM, and downstream tasks. As Table 2 already shows, each loss is not independent; AUDiff reduces EU, and EUDiff reduces AU as a side effect. If we simply summed the two losses, we would over-regularize both effects. It would also introduce two interacting trade-off weights, with no clear principled way to set them. In addition, combining the two objectives would add computational overhead.
> We therefore chose to report the clearer AU-only and EU-only variants in this first study, and view joint multi-objective optimization as an interesting direction for future work.
>
> Q4: What acceleration rate is used for the MRI experiments?
>
> Response: All MRI experiments in our paper use a 4 acceleration rate.

---

> ### Author Response · Authors · 2025-12-03
>
> Q5: Can you clarify the setups in Section 4.4? Section 4.4 is written in non-standard language for the medical imaging reconstruction community. I do not understand the three setups here. For instance, in setup 1, are you training a segmentation/classification network on zero-filled images (with artifacts)? Why would anyone do this? And why should this network generalize to "clean" reconstructions? Similar questions for the other two setups.
>
> Response: We appreciate the reviewer’s comment. We clarify that in all three settings in Section 4.4, the downstream classification task models are never trained on zero-filled images. Zero-filled images are used only as input conditions for the reconstruction model (HyperDM / AUDiff / EUDiff).
>
> Q6: It's hard to see any of the reconstructions in Fig. 2. Can you provide higher resolution versions with error maps?
>
> Response: Thanks for the suggestion. In Figure 2, our primary goal was to visualize the overall AU and EU patterns across methods, rather than fine-grained structural differences, which led us to use a compact layout. In the revised version, we will keep this overview figure in the main paper and add higher-resolution reconstructions with corresponding AU/EU maps and zoomed-in views in the appendix to make local reconstruction differences easier to inspect.
>
> Q7: Can you discuss how the method behaves when the protocol changes, e.g., a different sparse view in CT or a different undersampling rate in MRI?
>
> Response: Thanks for the question. In the current paper, we do not change the acquisition protocol between the training and test sets: all CT experiments use the same 45-view sparse sinogram, and all MRI experiments use the same 4x undersampling mask to generate zero-filled conditions. Thus, our empirical results support robustness within a fixed setting.
> Conceptually, the proposed noise-consistency and weight-consistency objectives are defined on the conditional input y (sinogram for CT or zero-filled image for MRI) and the sampled weights, and do not assume a specific number of views or acceleration factor. If the model is retrained or fine-tuned on data from a different sparse-view CT setup or a different MRI undersampling rate, the same training losses apply in the same way, and we would expect the same pattern we observe here: reduced AU/EU without degrading PSNR/SSIM and downstream performance.
>
> Q8: How does the quality of DMs defined through (z) sampled from the BHN framework compare to a single DM trained directly with the appropriate loss?
>
> Response: We appreciate the question. Conceptually, a single DM trained with the appropriate loss corresponds to one fixed set of weights and produces a single reconstructed image per input. In the BHN framework, sampling different 𝑧 yields multiple plausible weight sets (multiple experts) and therefore multiple reconstructed images. Aggregating these reconstructed images gives a reconstructed image that is more stable and less sensitive to any single badly tuned weight configuration, so the mean BHN-based DM reconstruction is high quality compared to a single DM and more robust in regions where a single model can deviate significantly from the ground truth.

---

### Official Review · Reviewer_BRYX · 2025-10-31

**Soundness:** 3
**Presentation:** 2
**Contribution:** 2
**Rating:** 4
**Confidence:** 4

**Summary:**

The presented approach builds on HyperDM and presents a diffusion-based framework for medical image reconstruction that aims to reduce rather than simply estimate uncertainty. It combines a pre-trained diffusion model with a Bayesian hypernetwork that generates the model’s weights and introduces two training objectives addressing different uncertainty sources. The noise consistency loss makes reconstructions stable under small input perturbations, reducing uncertainty caused by measurement noise, while the weight consistency loss encourages agreement among reconstructions from different sampled weight sets, reducing model-related uncertainty. Only the hypernetwork is optimized while the diffusion model remains fixed. Experiments on computed tomography and magnetic resonance imaging show that both uncertainty types are reduced without loss of image quality, and that downstream tasks such as lung nodule segmentation and pathology classification improve.

**Strengths:**

The clear disentanglement and targeted reduction of aleatoric and epistemic uncertainty are novel and interesting. The paper is in most parts clearly written, easy to follow, and provides sufficient context and explanation of the prior work it builds upon. The proposed objectives are simple yet well motivated, offering an intuitive and principled way to make uncertainty a direct optimization goal rather than a secondary outcome of model estimation.

**Weaknesses:**

My first major concern lies in how the hypernetwork is trained. As I understand it, the hyper network parameters must be updated by unrolling the full diffusion process and backpropagating through it, which makes the approach computationally expensive and difficult to scale. A second major concern is the poor reconstruction performance observed on the fastMRI dataset. It remains unclear why the base diffusion model performs so poorly, is this a consequence of using a lightweight model for feasibility, or would the method fail to work with a higher-quality backbone? In addition, the authors model measurement noise as simple Gaussian perturbations, which does not reflect the actual acquisition process in many medical imaging modalities. In fastMRI, for example, undersampling is typically performed using a random k-space mask rather than additive Gaussian noise. It would be important to clarify whether the proposed approach could handle such structured, non-Gaussian noise.

**Questions:**

1. How is the hypernetwork trained in practice? Does it require unrolling the entire diffusion process and backpropagating through it, and if so, how does this affect scalability?

2. What explains the poor reconstruction quality on the fastMRI dataset? Is the low performance due to a simplified or lightweight base model chosen for feasibility, or would the method struggle to work with a stronger backbone?

3. Why is measurement noise modeled as Gaussian when many medical imaging modalities, such as fastMRI, involve structured or non-Gaussian noise (e.g., random k-space masks)?

4. Could the proposed approach handle more realistic noise models or acquisition processes beyond additive Gaussian noise?

---

> ### Author Response · Authors · 2025-12-03
>
> We thank the reviewer for the detailed and constructive feedback and for highlighting both the strengths and weaknesses of our work. Below, we address each question in turn.
>
> Q1: How is the hypernetwork trained in practice? Does it require unrolling the entire diffusion process and backpropagating through it, and if so, how does this affect scalability?
>
> Response: Thanks for the question. The hypernetwork is trained with the standard single-step diffusion objective. For each mini-batch, we sample a single random timestep t, apply the forward diffusion process to the ground truth image to obtain the noisy input xt, use the hypernetwork to produce the U-Net weights, run one forward pass, compute the Bayesian hypernetwork loss, and backpropagate once. We do not unroll the entire diffusion trajectory or backpropagate through multiple denoising steps. As a result, the training cost and memory usage are similar to a standard diffusion model.
>
> Q2: What explains the poor reconstruction quality on the fastMRI dataset? Is the low performance due to a simplified or lightweight base model chosen for feasibility, or would the method struggle to work with a stronger backbone?
>
> Response: Thanks for the questions. On fastMRI, the lower PSNR/SSIM is mainly due to our simplified backbone and conditioning, not a limitation of the proposed objectives. We deliberately use a generic 2D U-Net diffusion model with a lightweight Bayesian hypernetwork and zero-filled images as the only condition. Within this fixed HyperDM-style setup, AUDiff and EUDiff consistently reduce AU/EU and improve PSNR/SSIM over the HyperDM baseline (Table 2). Applying the same uncertainty-reduction losses to a stronger backbone is conceptually straightforward but would require additional engineering for a larger, more complex model, which we leave for future work.
>
> Q3: Why is measurement noise modeled as Gaussian when many medical imaging modalities, such as fastMRI, involve structured or non-Gaussian noise (e.g., random k-space masks)?
>
> Response: We appreciate the question. In our MRI setup, the “structured” corruption (the 4x equispaced k-space mask) is not treated as noise. We first apply that mask in k-space, zero-fill the missing lines, and take the inverse FFT to obtain the aliased zero-filled image. This zero-filled image is the condition given to the diffusion model. For AUDiff training, we then add small zero-mean Gaussian noise directly to this zero-filled image to create slightly perturbed versions of the same condition. Modeling this additional term as Gaussian is a standard simplification in MRI. The k-space mask captures the structured part of the acquisition, and the Gaussian noise is only used to represent small additional measurement perturbations around that fixed 4x undersampled setup.
>
> Q4: Could the proposed approach handle more realistic noise models or acquisition processes beyond additive Gaussian noise?
>
> Response: Thanks for the question. Our current AU reduction uses additive Gaussian noise on the condition 𝑦 for simplicity, but this is not a fundamental restriction. The AU loss only requires multiple perturbed versions of the condition, and the EU loss only requires multiple sampled weights from the Bayesian hypernetwork. As long as we can generate perturbed conditions and sampled different weights, the same AU and EU objectives can be used without changing their formulation.

---

### Official Review · Reviewer_mf6T · 2025-11-03

**Soundness:** 2
**Presentation:** 2
**Contribution:** 2
**Rating:** 4
**Confidence:** 3

**Summary:**

The paper extends diffusion-based reconstruction by introducing a Bayesian hypernetwork (BHN) that explicitly reduces aleatoric (AU) and epistemic (EU) uncertainty. Two objectives are trained separately: a noise-consistency loss to stabilize outputs under input perturbations (reducing AU), and a weight-consistency loss to enforce agreement across BHN-sampled weights (reducing EU). Experiments on sparse-view CT (LUNA16) and accelerated MRI (fastMRI) show significant AU/EU reduction while maintaining or improving PSNR/SSIM. Downstream segmentation and classification tasks also benefit from incorporating uncertainty maps.

**Strengths:**

Clear framing of uncertainty reduction, not just quantification.

Conceptually clean decomposition of AU and EU with separate training.

Strong empirical results on CT and MRI, including downstream clinical tasks.

Modular design—BHN applied on top of a diffusion model without retraining the base denoiser.

**Weaknesses:**

Reported AU/EU reductions are extreme (up to 8–9 orders of magnitude) without calibration or coverage analysis, raising concerns of posterior collapse.

Minimizing variance can trivially reduce uncertainty without ensuring realistic posterior spread.

CT experiments rely on simulated Gaussian noise rather than realistic Poisson or view-dependent models.

Computational cost (M×N sampling) is high and unreported.

No test of whether a single joint AU/EU objective could balance both sources.

Lack of confidence intervals or significance testing for reconstruction and downstream metrics.

Evaluation limited to a single 2D diffusion backbone; unclear scalability to 3D or multi-coil MRI.

**Questions:**

How calibrated are AU and EU with respect to actual reconstruction error?

Does the weight-consistency term collapse the BHN posterior?

How sensitive are results to the number of weight and noise samples (M, N)?

How realistic are the CT measurement perturbations used for AU training?

What is the runtime and memory cost per reconstruction compared to baseline HyperDM?

How do improvements hold under out-of-distribution conditions (e.g., unseen masks or noise levels)?

---

> ### Author Response · Authors · 2025-12-03
>
> We thank the reviewer for the detailed and constructive feedback and for highlighting both the strengths and weaknesses of our work. Below, we address each question in turn.
>
> Q1: How calibrated are AU and EU with respect to actual reconstruction error?
>
> Response: Thanks for the question. AU and EU are not calibrated to the actual reconstruction error, because the uncertainty objectives in Equations 4 and 7 are defined independently of the reconstruction loss.
>
> Q2: Does the weight-consistency term collapse the BHN posterior?
>
> Response: Thanks for the question. The weight-consistency term is applied to the reconstructed images, not directly to the BHN’s weights. For a fixed input, it penalizes large differences between reconstructed images from different BHN-sampled weight sets. This makes the multiple sets of sampled weights correlated but it does not collapse the BHN posterior to a single set of weights. In our experiments, using a reasonable weight-consistency coefficient reduces epistemic variability in the predictions as intended without numerical instability or degenerate behavior.
>
> Q3: How sensitive are results to the number of weight and noise samples (M, N)?
>
> Response: We appreciate the question. In our setup, M and N control how many times we sample when computing the noise- and weight-consistency losses. During training, if M and N are too small (e.g., 1), these terms fluctuate more from batch to batch and act as a weak regularizer. Once we use small but reasonable values for M and N, training becomes stable, and PSNR/SSIM and AU/EU show the same qualitative trends. Increasing M and N further mainly increases computation without changing the overall behavior. At inference time, larger M and N yield smoother, more stable AU/EU maps, while very small values produce noisier maps. However, the location of high and low uncertainty remains similar. Overall, the results are not highly sensitive once M and N are set to reasonable values.
>
> Q4: How realistic are the CT measurement perturbations used for AU training?
>
> Response: Thanks for the question. For sparse-view CT, we simulate a 45-view sinogram and add small zero-mean Gaussian noise following [1], and this noisy sinogram is used as the condition for AUDiff training. In CT, although the raw measurements are Poisson, after preprocessing and log transform, they are often modeled as additive Gaussian noise for algorithm development. Our perturbations, therefore, do not capture the full Poisson–Gaussian physics, but they follow common sparse-view CT practice and provide a reasonable way to model small measurement variations around a fixed 45-view sinogram.
> [1] Chan, Matthew, Maria Molina, and Chris Metzler. "Estimating epistemic and aleatoric uncertainty with a single model." Advances in Neural Information Processing Systems 37 (2024): 109845-109870.
>
> Q5: What is the runtime and memory cost per reconstruction compared to baseline HyperDM?
>
> Response: We appreciate the question. At inference time, we measured the runtime and memory cost on an NVIDIA V100 32 GB GPU. HyperDM, AUDiff, and EUDiff use the same diffusion backbone, BHN, and sampling schedule (M = 10 weight samples, N = 100 trajectories). So, the number of forward passes and the GPU memory required per reconstruction are roughly the same between our proposed AUDiff, EUDiff, and the baseline HyperDM. In our setting (M x N = 1000 reconstructed images for a single slice), the average total runtime is approximately 98.17 seconds per input slice. The peak GPU memory usage during inference is approximately 1.23 GB.
>
> Q6: How do improvements hold under out-of-distribution conditions (e.g., unseen masks or noise levels)?
>
> Response: Thanks for the question. In this paper, we do not change the acquisition setup between training and test (CT uses the same 45-view setting; MRI uses the same 4x undersampling), so our reported improvements are in-distribution. We did not run experiments with unseen masks or noise levels, and therefore cannot claim robustness under such out-of-distribution conditions. Conceptually, the noise- and weight-consistency losses are designed to make predictions less sensitive to small input and weight perturbations, so they may help under moderate distribution shifts, but verifying this would require additional experiments.

---

### Meta-Review · Area_Chair_pnnS · 2026-01-09

**Summary:**

The paper proposes a diffusion-based medical image reconstruction framework with a Bayesian hypernetwork and two separate learning objectives to reduce aleatoric and epistemic uncertainty. The reviewers appreciated the paper's clear framing around unvertanty rediciton, the novelty of the AU and EU decomposed losses, and found that the paper, for the most part, is clearly written and easy to follow. However, the reviewers raised many major concerns, including the validity of reducing uncertainty (particularly in the EU), the theoretical grounding of the losses, issues with calibration, reliance on Gaussian noise, a limited evaluation scope (only 2D and no out-of-distribution conditions), and doubts about relevance to the medical imaging community.

**Reviewer Concerns:**

The rebuttal addressed many issues (including training without unfolding the diffusion process, running time and memory cost). However, some concerns remain unresolved, including the validity of the uncertainty reduction, the theoretical grounding of the losses, calibration issues, and the limited experimental scope.

**Reviewer Scores:**

Two reviewers (xJpW and PDT6) recommended rejecting the paper (score 2), with major concerns about the validity of reducing uncertainty and the theoretical grounding of the corresponding losses, as well as community relevance. Given that two reviewers recommend rejection and the remaining two rate the paper as "marginally below the acceptance threshold", I consider it unlikely that the overall evaluation would shift sufficiently toward acceptance, even if some reviewers were to increase their scores.

---

### Decision · Program_Chairs · 2026-01-26

Reject